# Ultrasound-guided medial branch of the superior laryngeal nerve block to reduce peri-operative opioids dosage and accelerate patient recovery

Qiuxiang Kang[1], Liang Wu[2], Yaohong Liu[3], Xu Zhang[1]*

1 Department of Anesthesiology, Ningbo Medical Centre Lihuili Hospital, Ningbo, China, 2 Department of Anesthesiology, Guilin Medical University Affiliated Hospital, Guilin, China, 3 Department of Anesthesiology, Hainan Hospital of Chinese PLA General Hospital, Hainan, China

* 450047830@qq.com

**Data Availability Statement:** All relevant data are within the paper and its Supporting Information files.

## Abstract

### Background

To explore whether the medial branch block of superior laryngeal nerve can reduce the stress response of patients undergoing intubation and further reduce the dosage of opioids.

### Methods

80 patients undergoing gynecological laparoscopic surgery were selected, and randomly divided into 4 groups. All patients in the experimental groups received bilateral internal branch of superior laryngeal nerve block and transversus abdominis plane block. But the dosage of sufentanil used for anesthesia induction in the group A, B, and C was 0.4, 0.2, and 0μg/kg, respectively. Group D do not underwent supralaryngeal nerve block and the dosage of sufentanil was 0.4μg/kg. The heart rate (HR) and mean arterial pressure(MAP) were recorded at the time of entering the operating room(T1), before intubation after induction(T2), immediately after intubation(T3), 5min after intubation(T4), before extubation(T5), immediately after extubation(T6), 5min after extubation(T7). We also recorded the stay time in the recovery room, the number of cases of postoperative sore throat, the number of cases of nausea and vomiting, the first intestinal exhaust time, the length of hospital stay after operation.

### Results

The HR of group A, C and D at T3 was significantly higher than that at T2(P < 0.01), while the HR of group B had no significant change. The HR of group A, C and D at T4 was lower than that at T3(P < 0.01), while the HR of group B had no obvious change. The HR of group C and D at T3 was significantly higher than that at T1 (P < 0.01). The MAP of group A and D at T4 was significantly lower than that at T1 (P<0.001). The first postoperative intestinal exhaust time in group A, B and C was significantly shorter than that in group D. The length of hospital stay after operation in group B and C was shorter than that in group D.

**Funding:** The author(s) received no specific funding for this work.

**Competing interests:** The authors have declared that no competing interests exist.

## Conclusions

Ultrasound-guided superior laryngeal nerve block combined with 0.2μg/kg sufentanil can reduce the intubation reaction, have better hemodynamic stability, reduce the first postoperative intestinal exhaust time and postoperative hospital stay, thereby accelerating the postoperative recovery of patients.

## Introduction

Perioperative pain management is one of the primary tasks of anesthesiologists. Poor pain control can have a range of adverse effects on patients. During hospitalization, poor perioperative pain control is associated with increased incidence of nausea and vomiting, impaired immune function, increased cardiac and pulmonary stress, delayed wound healing and prolonged hospitalization [1, 2]. Among patients with poor perioperative pain control, 10–30% develop chronic persistent postsurgical pain(CPPSP), which severely affects the patients' daily life and may lead to long-term use of opioids. Therefore, effective perioperative pain management can reduce patients' physiological and psychological stress responses, relieve pain, and promote healing and recovery [2, 3]. The main contents include: the anesthesiologist fully evaluates the patient's pain state before operation, and understands the use of analgesic drugs and the impact of pain on the patient's physiology and psychology; fully relieve pain during operation, minimize the surgical stimulation and stress reaction of patients, and on the basis of which rationally use drugs to speed up the recovery of patients; to provide patients with the optimal pain mode after operation, which can not only fully relieve pain, but also reduce drug-related complications without affecting the recovery of patients' normal physiological functions [4, 5].

The incidence of postoperative pain may be a bit higher than we thought. A recent meta-analysis showed that the incidence of moderate to severe postoperative pain in discharged patients ranged from 31% one day after discharge to 58% one to two weeks after discharge [6]. Therefore, perioperative pain management is still one of the main concerns of anesthesiologists.

In recent years, less opioids and multimodal analgesia have been widely concerned and strongly recommended in perioperative pain management [7,8]. Opioids can alleviate the pain of patients and stress response during intubation, but they also have many side effects, such as nausea and vomiting, respiratory depression, sleep disturbance and inhibition of gastrointestinal peristalsis [9]. Studies have shown that even a limited dose of opioids may lead to addiction; long-term opioids use increases the mortality of patients [10, 11]. Opioids may lead to tumor growth, angiogenesis and distant spread through transactivation of vascular endothelial growth factor (VEGF) receptors and increased expression of opioid receptors, which is closely related to the prognosis of cancer patients [12]. High doses of opioids carry a risk of long-term dependence, with 10 percent of patients continuing to use opioids for up to 90 days after undergoing a series of major and minor surgeries [2]. In many countries, opioids dependence has developed into a major public health problem, and in this context, strategies to minimize opioid use during the perioperative period while guaranteeing patient comfort and functional recovery are research priorities [13]. Therefore, anesthesiologists should adopt various analgesic measures to reduce the use of opioids during perioperative period to improve the recovery of patients.

Ultrasound-guided nerve block is one of the widely used perioperative analgesia methods. There are a variety of nerve block techniques to relieve incision pain in patients undergoing

different operations, such as abdominal transverse fascia block, quadratus lumborum block and paravertebral nerve block [14–16]. About 10% of patients experience sore throat after tracheal intubation under general anesthesia, which affects their comfort and satisfaction, but this has not been paid sufficient attention by surgeons and anesthesiologists [17]. It has been reported that the medial branch block of superior laryngeal nerve can relieve postoperative sore throat and reduce the incidence of postoperative cough and hoarseness in patients undergoing general anesthesia intubation after operation [18, 19]. Studies have also shown that superior laryngeal nerve block can increase the patient's tolerance, reduce hemodynamic fluctuations and shorten the time required for intubation when it is applied to difficult airway intubation guided by fiberoptic bronchoscope [20]. Our primary goal was to assess whether the medial branch block of the superior laryngeal nerve can reduce the stress response of patients undergoing general anesthesia during intubation. Our secondary goals were to assess whether the combination of other nerve blocks that can alleviate the incision pain can improve the perioperative pain of patients, thereby further reducing the dosage of opioids and speed up patient's postoperative recovery.

## Materials and methods

### General information

This study is a randomized controlled trial, approved by the Ethics Committee of Affiliated Hospital of Guilin Medical College (ethics batch number: YJSLL2021122) and registered in China Clinical Trial Center (http://www.chictr.org.cn)(registration number: ChiCTR2000038355 Principal investigator: Xu Zhang, Date of registration: September 21, 2020). All patients in this trial obtained the consent of themselves and their families and signed written informed consent forms. In this study, 80 patients undergoing gynecological laparoscopic surgery form May 9, 2021 to December 9, 2021 were selected. Inclusion criteria: (1) ASA grade: I ∼ II; (2) Age:18–65 years old; (3) Height: 145-185cm; (4) Weight: 45–80 kg; (5) Education level: above primary school; (6) Sign the informed consent form. Exclusion criteria: (1) Patients with full stomach and digestive tract obstruction; (2) Participated in clinical trials within 4 weeks before surgery; (3) Severe heart, lung, liver, kidney and other organ dysfunction; (4) Have a history of neurological and psychiatric disorders, and a history of long-term use of sedatives; (5) Patients with severe aphasia, auditory and visual impairment, or severe motor impairment who cannot cooperate with the examination; (6) Pregnant or lactating women; (7) Patients with motion sickness and allergy to ropivacaine; (8) Patients with thyroid-related diseases that affect the operation of superior laryngeal nerve block; (9) Patients who may have airway difficulties. Rejection criteria: (1) Blood loss during operation is greater than 500ml; (2) Enlarging the incision during operation; (3) The operation time is more than 4h; (4) Unpredictable patients with difficult airway. The CONSORT flow diagram of this study is as follows (Fig 1).

### Randomization and blinding

The enrolled patients were randomly divided into 4 groups by the administrator: A, B, C and D. Group A received ultrasound-guided bilateral superior laryngeal nerve medial branch block (2% lidocaine 3ml on each side) and ultrasound-guided bilateral transversus abdominis plane block (0.375% ropivacaine 15ml on each side). The dosage of sufentanil was 0.4μ g/kg during induction. The nerve block method of group B was the same as that of group A, but the induced dose of sufentanil was 0.2μ g/kg. The nerve block method of group C was the same as that of group A and B, but the induced dose of sufentanil was 0ug/kg; Group D was the control group, which underwent ultrasound-guided bilateral transverse abdominal fascia block

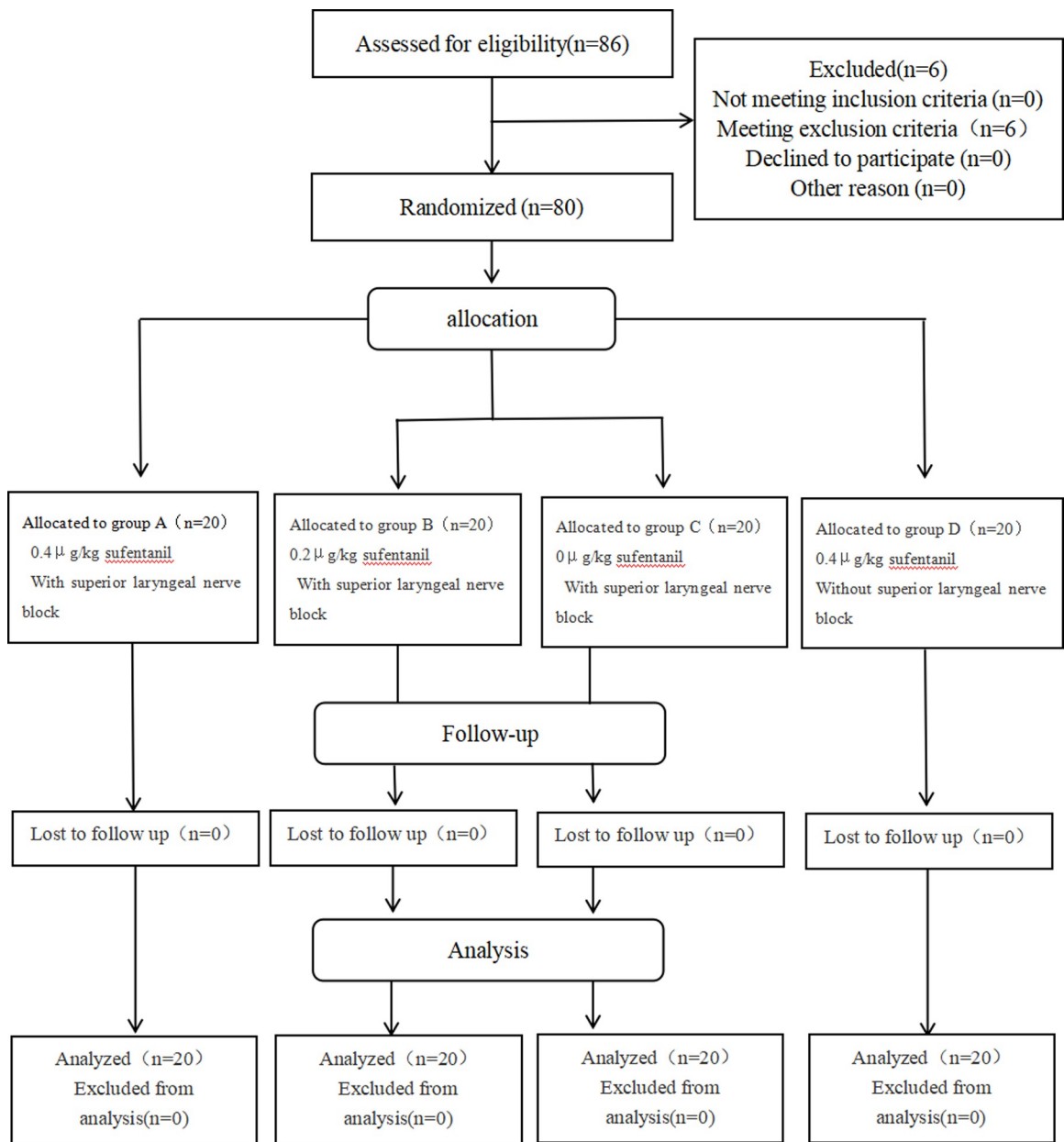

**Fig 1. The CONSORT flow diagram.**

(0.375% ropivacaine 15ml per side), but no supralaryngeal nerve block. The dosage of sufentanil was 0.4μg/kg during induction.

The administrator who don't take part in the treatment randomly divided the patients into groups, and generated the random numbers by Excel. Before the start of the experiment, the envelope concealed the random numbers was opened by the anesthesiologist who performed the punctuation and telephone the nurse to dispense the drugs. The anesthesiologist performed anesthesia induction and nerve block under the guidance of ultrasound. Patients in all groups were treated in the postanesthesia care unit and blinded to which treatment they would

receive. The outcome assessors, data collectors, and statisticians were blinded to group allocations during the study.

## Anesthesia methods

When the patients entered the operating room, midazolam 0.04mg/kg, propofol 1.8mg/kg, rocuronium 0.9mg/kg and sufentanil were used to induce anesthesia.(the sufentanil dose of experimental group A, B, and C were prepared by other researchers according to the numbers in the random number table, and the dose of control group D were 0.4μg/kg, uniformly diluted into 5ml). After induction, positive pressure ventilation with mask was performed, and at the same time, bilateral superior laryngeal nerve block and bilateral transverse abdominal fascia block were performed by a trained anesthesiologist. After the block is completed, endotracheal intubation can be performed by visual laryngoscope. Sevoflurane 1.5%, propofol 0.05mg/(kg. min), rocuronium 5ug/(kg.min) and remifentanil 0.06μg/(kg.min) were used for anesthesia maintenance.

**Ultrasound-guided medial branch block of superior laryngeal nerve.** To disinfect the neck skin, place a linear array probe (MicroTurboHFL38X, Sonosite Company, USA, probe frequency 6-13MHz) covered with ultrasonic protective sleeve transversely between hyoid bone and thyroid cartilage, and gradually slide outward, so that the hyperechoic thyroglossum can be seen in the image, the anechoic superior laryngeal artery can be found in the thyroglossum, and the internal branch of the superior laryngeal nerve has an external hyperechoic structure surrounding the internal hypoechoic structure. 3 ml of 2% lidocaine (Hebei Tiancheng Pharmaceutical Co., Ltd., batch number: H13022313) was injected into the periphery of the superior laryngeal nerve by acupuncture from the middle plane to block the medial branch of the superior laryngeal nerve. Complete the other nerve block in the same way at the same needle insertion point. Fig 2 for ultrasonic imaging.

**Transversus abdominis plane block guided by ultrasound.** The linear array probe covered with ultrasonic protective sleeve was placed between the costal margin and the anterior superior iliac spine. Under ultrasound, three layers of muscle structures could be seen, namely, external oblique abdominal muscle, internal oblique abdominal muscle and transverse abdominal muscle. 15ml of 0.375% ropivacaine (AstraZeneca AB, Sweden, batch number: LBYR) was injected between external oblique abdominal muscle and transverse abdominal muscle by in-plane acupuncture.

**Intraoperative management and treatment of special circumstances.** All patients used BIS monitoring to evaluate anesthesia depth in this experiment. The BIS value control target is between 40 and 60. The hemodynamic fluctuation control target of patients is within 20% of the basic level. During anesthesia, if the patient's BIS value and hemodynamic fluctuation range are not within the control target range, the anesthesiologist will analyze the related possible reasons and adjust the anesthesia depth and use vasoactive drugs, while making records.

## Observation indicators

Record the general data of patients, including their heart rate (HR) and mean arterial pressure (MAP) when they enter the operating room($T_1$), before intubation after induction($T_2$), immediately after intubation($T_3$)and 5min after intubation($T_4$). The anesthesia time, operation time, dosage of remifentanil and sufentanil were also recorded. The HR and MAP of patients before extubation($T_5$), immediately after extubation($T_6$) and 5min after extubation($T_7$), the patients' stay time in the recovery room, the incidence of postoperative sore throat, the frequency and duration of nausea and vomiting, and the intestinal exhaust time were recorded. The numeric rating scales (NRS) were recorded at 3h, 6h, 12h, 24h, 48h after operation, and

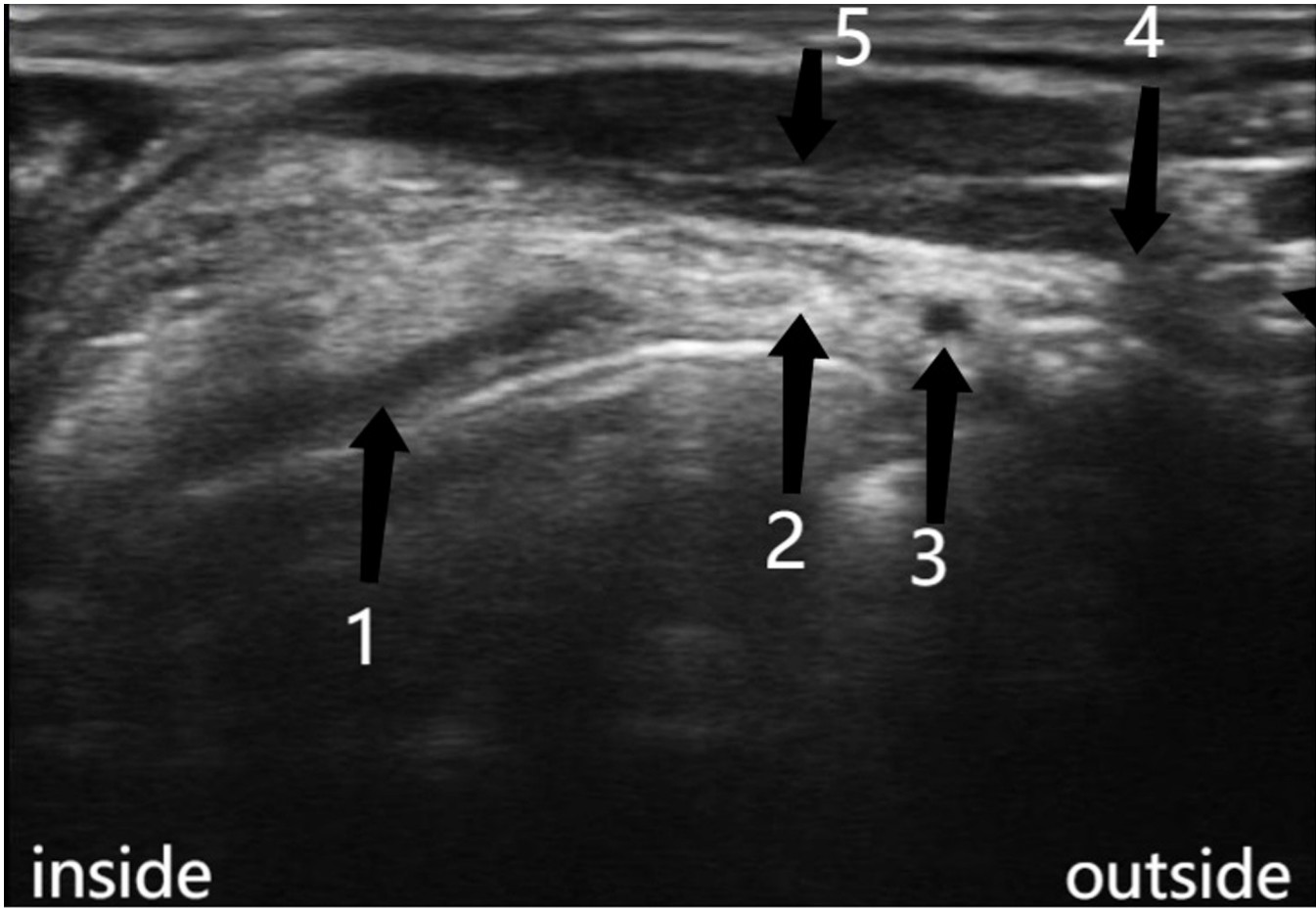

**Fig 2. Ultrasonic imaging of medial branch of superior laryngeal nerve.** 1: epiglottis; 2: medial branch of the superior laryngeal nerve group; 3: the superior laryngeal artery; 4: thyrohyoid muscle; 5: sternohyoid muscle.

whether analgesics were used in the ward and the kind and dosage were recorded. Record other special circumstances. Followed up the total hospitalization time and discharge time after operation.

## Sample size and statistical analysis

The largest difference was expected to be between groups A and C. Based on our pilot study, the heart rate change before and after intubation in group C was $11.7\pm2.3$ beats/minutes, and the change in group A was $19\pm6.6$ beats/minutes. Thus, the sample size was calculated with a power of 0.8 and alpha error of 0.05. At least 18 subjects per group were needed. Assuming a 10% dropout rate and to increase the credibility of this study, 80 patients were recruited (20 subjects for each group).

GraphPad Prism Version 8.0.1 statistical software is used for data analysis and graphic production. When the normal distribution measurement data are expressed as means±SD, one-way ANOVA is used for comparison between groups, and Sidak multiple comparison is used for pairwise comparison between groups. The measurement data were performed using chi-square test. The missing data were treated using the mean imputation method. $P<0.05$ indicates that the difference is statistically significant.

## Results

A total of 86 patients met the inclusion criteria and were enrolled in this study, of which 6 patients met the exclusion criteria and were excluded. There was no significant statistic difference in patient age, body mass index(BMI), American Association of Anesthesiologists(ASA) grade, anesthesia duration, operation duration, remifentanil dosage, blood loss and abdominal troca array ($P$>0.05)(Table 1).

Our study observed cardiovascular response during intubation and the postoperative indicators of patients and found that ultrasound-guided superior laryngeal nerve block binding with 0.2μg/kg sufentanil can reduce intubation response and improve hemodynamic stability, thereby shortening the first postoperative intestinal exhaust time and postoperative hospitalization time.

### Comparison of heart rate in different groups during intubation

There was no significant statistic difference in heart rate among group A, B, C and D at $T_1$ ($P$>0.05, Fig 3A). At $T_2$, the heart rate of group B and C was significantly higher than that of group A ($P$<0.01, Fig 3A), while the HR of group D was lower than that of group C($P$<0.05, Fig 3A). At $T_3$, the heart rate of group B and C was higher than that of group A ($P$<0.05, Fig 3A). At $T_4$, the heart rate of group B and C was higher than that of group A ($P$<0.05, Fig 3A), and the heart rate of group D was significantly lower than that of group C ($P$<0.01, Fig 3A). In group A, the HR at $T_3$ was significantly higher than that at T2 and $T_4$($P$ < 0.01, Fig 3B). There was no significant difference in heart rate in group B at the time of $T_1$,$T_2$,$T_3$ and $T_4$($P$>0.05, Fig 3B). In group C, the HR at $T_3$ was higher than that at $T_1$,$T_2$ and $T_4$($P$ < 0.05, Fig 3B), while in group D, the HR at $T_3$ was significantly higher than that at $T_1$,$T_2$ and $T_4$($P$ < 0.01, see Fig 3B). There was no significant difference in heart rate difference (△HR) between groups A, B, C and D ($P$>0.05, Fig 3C).

### Comparison of blood pressure in different groups during intubation

There was no significant difference in MAP among group A, B, C and D at T1($P$>0.05). At $T_2$, the MAP of group A was lower than that of group B ($P$<0.05, Fig 4A), while that of group C was significantly higher than that of group A and D ($P$<0.01, Fig 4A). At $T_3$, the MAP of group C was higher than that of group A, B and D ($P$<0.05, Fig 4A). At $T_4$, the MAP of group

**Table 1. Comparison of general data of patients in each group.**

| project | A group | B group | C group | D group |
|---|---|---|---|---|
| Age | 42.80±9.04 | 44.16±10.00 | 40.90±6.84 | 44.6±7.89 |
| BMI | 23.00±2.19 | 22.07±3.17 | 22.00±3.08 | 23.81±2.88 |
| ASA grading (/) case numbers | 13/7 | 10/10 | 14/6 | 12/8 |
| Anesthesia duration | 158.90±58.47 | 144.50±41.37 | 138.80±36.60 | 150.00±52.25 |
| Operation duration | 128.10±55.15 | 118.05±40.60 | 116.40±51.94 | 123.70±50.93 |
| Remifentanil dosage (μg/kg/min) | 0.070±0.021 | 0.073±0.015 | 0.071±0.026 | 0.079±0.018 |
| hemorrhage volume | 68.50±34.68 | 47.50±22.45 | 55.50±28.37 | 69.50±53.95 |
| Troca number of abdominal cavity | 4 | 4 | 4 | 4 |

Group A, Group B and Group C were all subjected to superior laryngeal nerve block, and the dosage of sufentanil was 0.4μg/kg, 0.2μg/kg and 0 μg/kg respectively. In group D, there was no superior laryngeal nerve block, and the dosage of sufentanil was 0.4 μg/kg.

BMI: body mass index

ASA: American Association of Anesthesiologists.

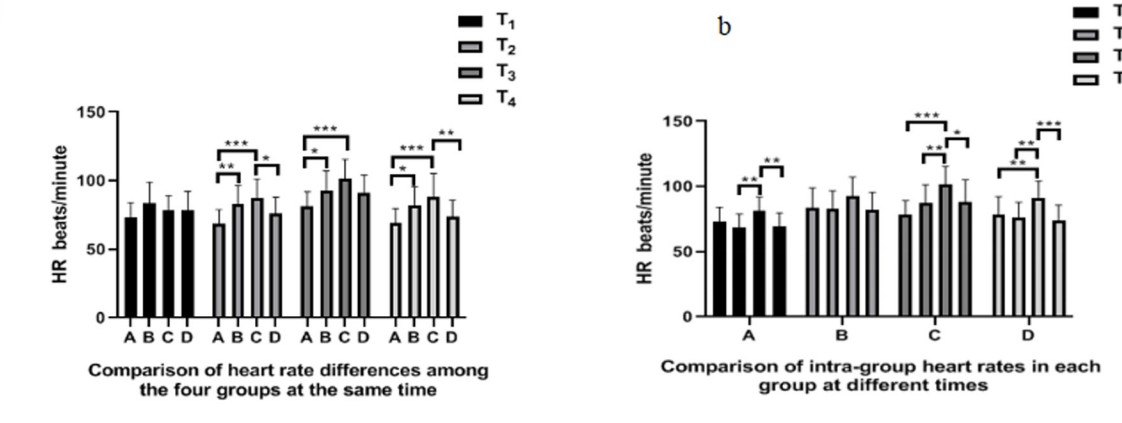

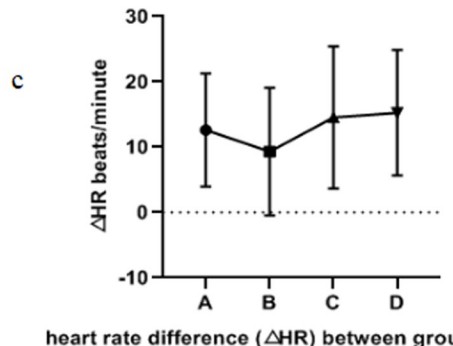

**Fig 3. Analysis of heart rate difference of each group during endotracheal intubation.** a. Comparison of heart rate among groups at different times. b. Comparison of intra-group heart rates in each group at different times. c.Comparison of heart rate difference ($\triangle$HR) among groups before and immediately after intubation. *: $P<0.05$, * *: $P<0.01$, * * *: $P<0.001$.

C was higher than that of group A and D ($P<0.05$, Fig 4A). The MAP of group A at $T_3$ was higher than$T_2$($P<0.001$,Fig 4B), and the MAP of group A at $T_2$ and $T_4$ was significantly lower than that at $T_1$($P<0.001$, Fig 4B). In group B, the MAP at $T_2$ was significantly lower than that at $T_1$ and $T_3$($P < 0.001$, Fig 4B). In group C, MAP at $T_3$ was significantly higher than that at $T_1$,$T_2$ and $T_4$($P < 0.01$, Fig 4B). In group D, the MAP at $T_1$ and $T_3$ was significantly higher than that at $T_2$($P<0.05$, Fig 4B). The difference of the average arterial pressure ($\triangle$MAP) between groups A, B, C and D was not statistically significant ($P>0.05$, Fig 4C).

## Comparison of heart rate and blood pressure in different groups during extubation

There was no statistically significant difference in MAP among group A, B, C and D at $T_5$,$T_6$ and $T_7$($P>0.05$). There was no significant difference in HR among group A, B, C and D at $T_5$, but there was differences in HR at $T_6$ and $T_7$($P<0.05$, Table 2).

## Comparison of postoperative indicators in different groups

There was no statistically significant difference among group A, B, C and D in the length of stay in Postanesthesia care unit(PACU), NRS scores at 3h, 6h, 12h, 24h and 48h after operation

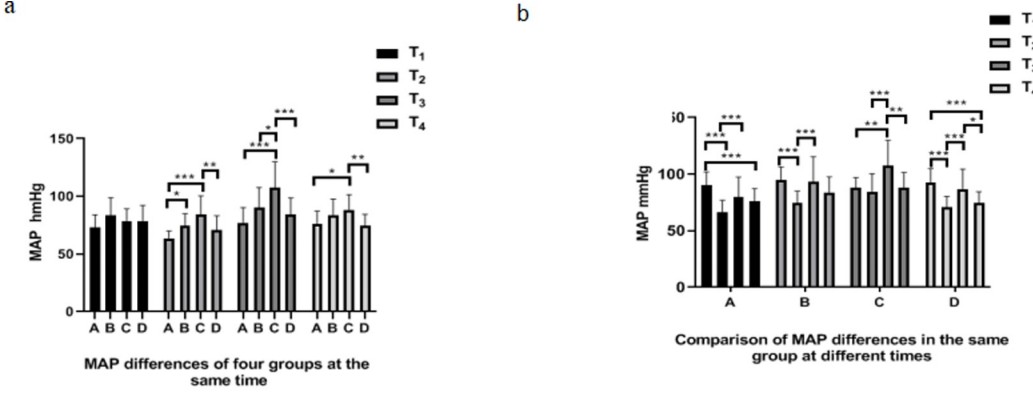

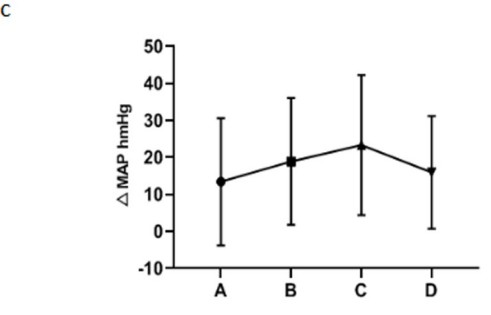

**Fig 4. Analysis of MAP difference among groups during endotracheal intubation.** a. Comparative analysis chart of MAP differences of each group at the same time point. b. MAP difference analysis chart of the same group at different time points. c. Comparative analysis of mean arterial pressure difference ($\triangle$MAP) among groups before and after intubation. *:$P<0.05$, * * :$P<0.01$,* * *: $P<0.001$.

**Table 2. Intra-group comparison and inter-group comparison of heart rate and blood pressure in each group during tracheal extubation.**

| Index | Group A | Group B | Group C | Group D | P value |
|---|---|---|---|---|---|
| **MAP ($T_1$)** | 90.10±12.00 | 94.78±11.69 | 88.33±8.70 | 92.87±12.05 | 0.227 |
| **MAP ($T_5$)** | 92.00±17.01 | 90.55±15.86 | 93.85±14.19 | 101.68±18.28 | 0.149 |
| **MAP ($T_6$)** | 94.97±15.55 | 96.08±17.73 | 95.63±16.52 | 100.92±17.74 | 0.672 |
| **MAP ($T_7$)** | 93.85±14.74 | 95.00±12.99 | 88.82±13.22 | 96.07±12.95 | 0.342 |
| ***P*value** | 0.746 | 0.653 | 0.231 | 0.236 | |
| **HR($T_1$)** | 73.20±10.84 | 83.80±15.16 | 78.50±10.55 | 78.40±13.80 | 0.083 |
| **HR ($T_5$)** | 78.00±14.90 | 75.50±15.28 | 77.70±10.88 | 85.65±19.70 | 0.188 |
| **HR ($T_6$)** | 84.00±14.39 | 83.50±12.88 | 78.55±10.38[a] | 91.40±14.57 | 0.027 |
| **HR ($T_7$)** | 80.20±12.87 | 74.30±19.26 | 74.15±9.87[a] | 85.00±11.09 | 0.047 |
| ***P* value** | 0.089 | 0.112 | 0.497 | 0.069 | |

Group A, Group B and Group C were all subjected to superior laryngeal nerve block, and the dosage of sufentanil was 0.4μg/kg, 0.2μg/kg and 0 μ g/kg respectively. In group D, there was no superior laryngeal nerve block, and the dosage of sufentanil was 0.4 μg/kg. $P^a<0.05$ indicates that the difference is statistically significant compared with group D

MAP: the average arterial pressure, HR: Heart Rate.

and the length of hospitalization after operation ($P > 0.05$, Table 3). The intestinal exhaust time of patients in group B and C was shorter than that in group D, and the difference was statistically significant ($P < 0.05$, Fig 5A). The postoperative hospital stay in group B and C was shorter than that in group D, and the difference was statistically significant ($P < 0.05$, Fig 5B).

## Discussion

### Role of multimodal analgesia in ERAS

ERAS aims to reduce the length of hospital stay, reduce medical costs and improve treatment satisfaction by adopting a series of interventions during perioperative management to avoid postoperative physiological and psychological adverse reactions of patients, and the ERAS programs have been successful in the past two decades [21, 22]. The ERAS protocol encourages multimodal analgesia and recommends the use of acetaminophen, non-steroidal anti-inflammatory drugs (NSAIDs), and gabapentin to limit opioids use to prevent postoperative ileus [23]. Good perioperative pain management can help eliminate acute pain caused by surgical trauma, improve patients' sleep, enhance immune function, make patients dare to cough and excrete sputum, get out of bed early, reduce the incidence of postoperative complications such as pulmonary infection, venous embolism of lower limbs, intestinal adhesion, and shorten the length of hospital stay and reduce the hospitalization costs [24, 25].

### Clinical role and advantages of superior laryngeal nerve block guided by ultrasound

The superior laryngeal nerve originates from the inferior ganglion of the vagus nerve on both sides of the neck and is divided into medial and lateral branches. The medial branch is the sensory nerve, which passes through the thyroglossal membrane and distributes in the laryngeal mucosa, epiglottis and tongue root above the glottis fissure. It is also the afferent nerve branch of cough and laryngeal spasm reflex. The lateral branch is the motor nerve, which innervates cricothyroid muscle and pharyngeal constrictor muscle [26]. Inoue Shiori et al. reported a case of acute epiglottitis in which only superior laryngeal nerve block could relieve the patient's discomfort and prevent serious complications caused by laryngeal obstruction during conscious tracheal intubation with video laryngoscope [27]. In addition, some studies have shown that superior laryngeal nerve block can reduce the cough score during bronchoscopy and the incidence of hypoxemia. Compared with the traditional local anesthesia combined with atomized

**Table 3. Comparison of postoperative indexes of different groups.**

| Postoperative index | Group A | Group B | Group C | Group D | P value |
|---|---|---|---|---|---|
| PACU stay time | 32.40±14.37 | 29.83±12.85 | 32.11±10.70 | 41.65±15.69 | 0.051 |
| NRS score at postoperative 3h | 2.30±1.34 | 3.30±1.86 | 2.30±1.30 | 2.30±1.49 | 0.098 |
| NRS score at postoperative 6h | 2.20±1.28 | 3.15±1.76 | 2.35±1.30 | 2.35±1.46 | 0.167 |
| NRS score at postoperative 12h | 2.25±1.25 | 2.85±1.73 | 1.95±1.10 | 2.10±1.25 | 0.174 |
| NRS score at postoperative 24h | 1.80±1.00 | 2.40±1.47 | 1.75±1.16 | 1.90±1.17 | 0.311 |
| NRS score at postoperative 48h | 1.30±0.86 | 1.65±1.30 | 1.30±0.98 | 1.50±1.00 | 0.667 |
| Total length of hospital stay(Day) | 8.45±2.78 | 9.50±3.22 | 8.55±2.37 | 10.95±4.44 | 0.069 |
| Sore throat cases number (occurrence/non-occurrence) | 7/13 | 3/17 | 8/12 | 5/15 | 0.308 |
| Nausea and vomiting cases number (occurrence/non-occurrence) | 8/12 | 3/17 | 5/15 | 6/14 | 0.353 |

Group A, Group B and Group C were all subjected to superior laryngeal nerve block, and the dosage of sufentanil was 0.4μg/kg, 0.2μg/kg and 0 μg/kg respectively. In group D, there was no superior laryngeal nerve block, and the dosage of sufentanil was 0.4 μg/kg. NRS score: pain intensity score.

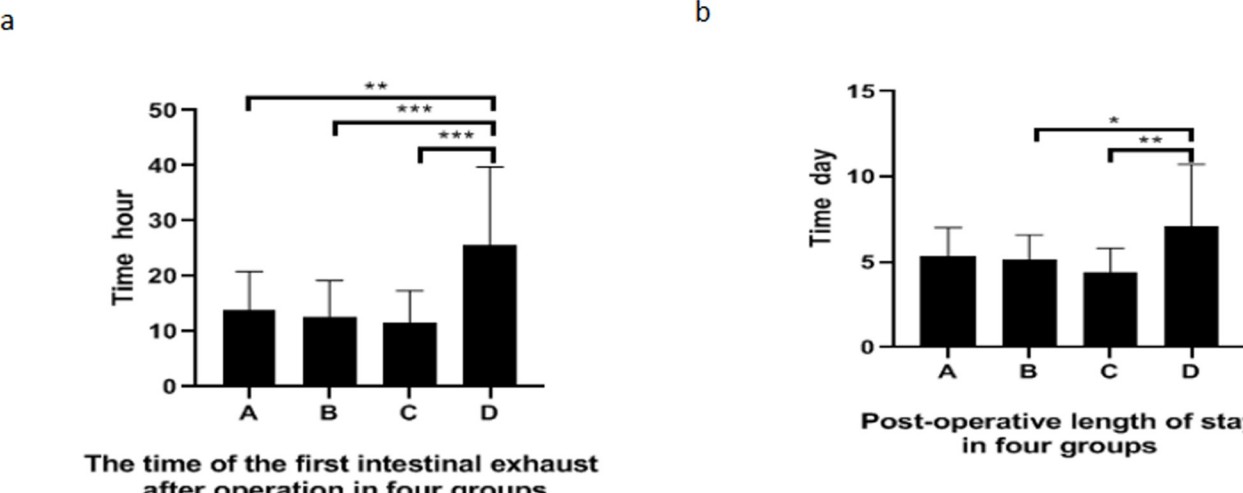

**Fig 5. Comparison of postoperative indicators in different groups.** a. the first time of intestinal exhaust in each group, b. the length of hospitalization after operation in each group. *: $P<0.05$, **: $P<0.01$, ***: $P<0.001$.

hormone inhalation, it has a better therapeutic effect on postoperative sore throat [17]. Therefore, superior laryngeal nerve block can inhibit the stress response during tracheal intubation. Clinically, the commonly used methods to reduce intubation reaction also include injection of local anesthetic through cricothyroid membrane and local surface anesthesia at throat mucosa, epiglottis and tongue root. Thyrocricocentesis may lead to tracheal wall bleeding and hematoma formation, which may lead to airway obstruction, and may also cause severe cough of patients, resulting in adverse experience [28]. In order to achieve a good local anesthetic effect on laryngeal mucosa, epiglottis and tongue root, a large number of local anesthetics are needed, which increases the risk of local anesthetic poisoning [29]. Ultrasound guided superior laryngeal nerve block with clear location and less local soft tissue damage will not cause tracheal wall bleeding and hematoma formation, and the use of local anesthetics is limited. It can be performed with sufficient sedation and analgesia, making patients feel better [30, 31]. In the current research reports, superior laryngeal nerve block is mainly used in the treatment of difficult airway, and the treatment of postoperative sore throat and neurogenic cough [32, 33]. Although it has been reported that the application of superior laryngeal nerve block alone can meet the needs of conscious tracheal intubation, the superior laryngeal nerve does not receive sensory afferent from the mucosa below the glottis. Therefore, it is reasonable to assume that the application of superior laryngeal nerve block alone may also cause some tracheal intubation reactions. However, whether ultrasound-guided superior laryngeal nerve block combined with some sedation and analgesia can fully inhibit tracheal intubation reaction during anesthesia induction and whether the combined application of regional block in other parts can further reduce the use of opioids has not been reported. In order to find the best dosage of analgesic drugs, we designed this study. We found that ultrasound-guided superior laryngeal nerve block combined with intravenous sufentanil of 0.2μg/kg during anesthesia induction can effectively inhibit tracheal intubation reaction and reduce hemodynamic fluctuation, which is obviously superior to the other three groups, and may be the best choice. It provides evidence for the application of superior laryngeal nerve block in anesthesia induction of some special patients (such as difficult airway, severe cardiovascular diseases and elderly patients,

etc.), which can increase the comfort and safety of patients and reduce the incidence of complications.

## Ultrasound-guided superior laryngeal nerve block combined with intravenous sufentanil of 0.2μg/kg can effectively inhibit tracheal intubation reaction

In this study, by comparing the heart rate changes of four groups of patients during intubation, we found that the heart rate of patients in group A, C and D at T3 was significantly higher than that at T2,while that of patients in group B at T2 and T3 had no statistical difference. Surprisingly, the changes of heart rate before and after tracheal intubation in group A combined with intravenous 0.4μg/kg sufentanil were more obvious than those in group B combined with intravenous 0.2μg/kg sufentanil. We analyzed the changes of heart rate at the time of entering the room and before intubation (after using sufentanil),and found that there was no significant change in these four groups (Fig 3B). After further analysis, we found that there was no significant difference in heart rate among the four groups (Fig 3A). But before intubation, the heart rate of group A was significantly lower than that of group B and C, and that of group D was lower than that of group C (Fig 3A). We also compared the difference of heart rate among groups before and after intubation and found that there was no significant statistical significance among the four groups (Fig 3C), but the average heart rate difference of Group B was lower than that of other groups (the absence of statistical difference may be related to insufficient sample size).The inhibitory effect of sufentanil on hemodynamics has been confirmed by many studies [34–36]. We believe that the larger dose of sufentanil has a more obvious inhibitory effect on the heart rate, thus making the heart rate change before and after intubation greater. Therefore, ultrasound-guided superior laryngeal nerve block combined with 0.2μg/kg sufentanil can better reduce the change of heart rate before and after tracheal intubation than the other three groups. In order to further compare the stability of heart rate of patients in each group after tracheal intubation, we recorded the heart rate of patients in each group 5 minutes after tracheal intubation. We found that there was no statistical difference in heart rate of patients in group B at the time of T1,T2,T3 and T4. In group A, C and D, the heart rate at T4 was lower than that at T3. The heart rate at T3 in group C and group D was significantly higher than that at T1. In group A, there was no significant difference between the heart rate at T3 and T1. The heart rate stability of patients in group B during endotracheal intubation was better than that of patientsthe in other three groups.

We also compared the mean arterial pressure(MAP) of four groups of patients during tracheal intubation. The MAP of four groups of patients at T3 was significantly higher than that at T2(Fig 4B). Here, we can't directly see the advantages and disadvantages of the four schemes. After further analysis, we found that there was no significant difference in the MAP of the four groups at T1(Fig 4A).The MAP at T2 in group C was no different from that at T1, while the MAP at T2 in group A, B and D was lower than that at T1(Fig 4B). At T2,The MAP of group A is lower than that of group B and C and the MAP of group D was significantly lower than that of group C (Fig 4A).It can be seen that the superior laryngeal nerve has no circulatory inhibition effect, but the higher the dose of sufentanil, the more obvious the circulatory inhibition effect of patients. This is consistent with previous research [17, 37]. By comparing the difference between MAP at T3 and T1, we found that there was no significant difference among groups A, B and D, but the MAP at T3 in group C was significantly higher than that at T1. From this, it can be seen that during endotracheal intubation, superior laryngeal nerve block alone may not be suitable for the elderly and patients with cardiovascular and cerebrovascular diseases, but intravenous combined application of sufentanil above 0.2μg/kg

can better meet clinical needs. We also compared the difference of MAP among the four groups before and after intubation and found that there was no significant difference among the four groups (Fig 4C). We further compared the stability of MAP in each group after intubation and found that the MAP in group C and group D at 5min after intubation (T4) was lower than that immediately after intubation (T3). However, there was no significant difference in MAP in group A and group B at T4 and T3. It can be seen that neither group C with superior laryngeal nerve block alone nor group D with sufentanil alone had stable hemodynamics.

By comprehensively analyzing the fluctuation of heart rate and blood pressure during endotracheal intubation, we found that ultrasound-guided superior laryngeal nerve block combined with intravenous application of 0.2μg/kg sufentanil had obvious advantages over other groups in inhibiting endotracheal intubation response and maintaining hemodynamic stability during endotracheal intubation.

## Combined with the medial branch block of superior laryngeal nerve during induction of general anesthesia can shorten the first postoperative intestinal exhaust time and postoperative hospitalization time

In this study, by comparing the heart rate and MAP at time of T5, T6 and T7, it was found that the heart rate of group C was lower than that of group D at T6 and T7 (Table 2). However, there was no statistical difference in heart rate and MAP during extubation among the other groups (Table 2). After careful analysis of the data, we found that the average heart rate and MAP of groups A, B and C during extubation were lower than those of group D (Table 2). Therefore, we think that blocking the medial branch of the superior laryngeal nerve may reduce the extubation reaction. However, there is no statistical difference among group A, B and D(P>0.05), which may be related to insufficient sample size. It is necessary to increase the sample size to determine whether the blockage of the medial branch of the superior laryngeal nerve can reduce the extubation reaction. Yin Bao et al. found that block of the medial branch of superior laryngeal nerve can reduce the cardiovascular response during extubation, which is different from our findings [38]. The local anesthetic they used in their study was ropivacaine, which works longer than lidocaine. So another reason for our findings may have to do with the duration of action of lidocaine.

Interestingly, by comparing the first postoperative intestinal exhaust time and postoperative hospitalization time of four groups of patients, we found that the first postoperative intestinal exhaust time of group A, B and C was significantly shorter than that of group D (Fig 5A). The length of postoperative hospital stay in group B and C was shorter than that in group D (Fig 5B). Here, we can't explain the difference of the first intestinal exhaust time in each group by the difference of opioids application. This indicates that these differences are caused by the blockage of the medial branch of the superior laryngeal nerve. However, it is not clear why the medial branch block of the superior laryngeal nerve can shorten the first postoperative intestinal exhaust time. We speculate that this may be related to the blocking of the medial branch of the superior laryngeal nerve, which alters the function of the vagus nerve and thus affects the peristalsis of the digestive tract. Lang IM et al. found that cutting off the superior laryngeal nerve did not change the esophageal motor function [39]. The related research on the influence of superior laryngeal nerve on gastrointestinal function has not been reported.

We also compared the PACU stay time, postoperative pain scores at 3h, 6h, 12h, 24h and 48h, the total length of stay, the incidence of sore throat, and the incidence of nausea and vomiting in each group, then found that there was no significant difference (Table 3). Therefore, the blocking of the medial branch of the superior laryngeal nerve may have no significant effect

on postoperative sore throat and postoperative nausea and vomiting. Zhipeng Li et al. found that superior laryngeal nerve block can reduce the score of postoperative sore throat, but the duration is not long [17]. This is basically consistent with our research results. This may also be related to the short duration of action of lidocaine, as our findings differ from those of Yin Bao et al using ropivacaine [38].

## Limitations

This study also has some shortcomings. First of all, some data need to be recorded in time, it is difficult for data collectors to avoid the operation process during the experiment. Secondly, this is a single-center study with a small number of included cases. Third, patients were only followed up within 48h, and no long-term follow-up was conducted to understand the long-term prognosis of patients. Finally, this study did not examine the content of catecholamine hormones in the blood of patients at different time points, which would be more convincing if this indicator was included in the observational indicators.

## Conclusions

Ultrasound-guided medial branch block of superior laryngeal nerve combined with intravenous use of 0.2μg/kg sufentanil during tracheal intubation in general anesthesia surgery has better hemodynamic stability, reduces the need for opioids during intubation. Ultrasound-guided SLNB combined with TFPB can reduce perioperative opioids use, shorten PACU residence time and intestinal first exhaust time of patients.This study did not clarify how the medial branch block of the superior laryngeal nerve can shorten the first postoperative intestinal exhaust time and postoperative hospitalization time, which needs further study. Whether special populations (such as patients with airway difficulties, severe cardiovascular diseases, and elderly patients, etc.) can benefit from this protocol is also worthy further study.

## Supporting information

**S1 Checklist. Reporting checklist for randomised trial.**
(DOCX)

**S1 Dataset.**
(XLS)

**S1 Protocol.**
(DOCX)

## Author Contributions

**Conceptualization:** Xu Zhang.

**Data curation:** Liang Wu.

**Formal analysis:** Yaohong Liu.

**Investigation:** Liang Wu.

**Supervision:** Xu Zhang.

**Validation:** Yaohong Liu.

**Writing – original draft:** Qiuxiang Kang.

**Writing – review & editing:** Xu Zhang.

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
