## [Decision Letter · Decision Letter 0]

2 Aug 2023

PONE-D-23-20292Ultrasound-guided medial branch of the superior laryngeal nerve block to reduce peri-operative opioids dosage and accelerate patient recoveryPLOS ONE

Dear Dr. Zhang,

Thank you for submitting your manuscript to PLOS ONE. After careful consideration, we feel that it has merit but does not fully meet PLOS ONE’s publication criteria as it currently stands. Therefore, we invite you to submit a revised version of the manuscript that addresses the points raised during the review process.

The introduction needs more context, including an overview of perioperative pain management importance, postoperative pain prevalence, and the significance of reducing opioid usage, with supporting statistics or evidence for multimodal analgesia. Define all abbreviations, particularly in the abstract and introduction. Clarify the sample size justification, whether based on primary outcomes. Expand the discussion to interpret results in the context of existing literature, discuss implications for perioperative pain management, study limitations, and future research directions. Provide a concise summary of key findings and their implications. Proofread the paper to improve overall quality by addressing grammar and typographical errors.==============================

We look forward to receiving your revised manuscript.

Kind regards,

Lalit Gupta

Academic Editor

PLOS ONE

Note: HTML markup is below. Please do not edit.]

Reviewers' comments:

Reviewer's Responses to Questions

**Comments to the Author**

1. Is the manuscript technically sound, and do the data support the conclusions?

Reviewer #1: Yes

Reviewer #2: Yes

2. Has the statistical analysis been performed appropriately and rigorously? 

Reviewer #1: Yes

Reviewer #2: I Don't Know

3. Have the authors made all data underlying the findings in their manuscript fully available?

Reviewer #1: Yes

Reviewer #2: Yes

4. Is the manuscript presented in an intelligible fashion and written in standard English?

Reviewer #1: Yes

Reviewer #2: Yes

5. Review Comments to the Author

Reviewer #1: Hi dear chief editor

I studied your valuable manuscript by title"Ultrasound-guided medial branch of the superior laryngeal nerve block to reduce peri-

operative opioids dosage and accelerate patient recovery" it have not Keyword, so write it on manuscript.

Best regard

The reviewer

Reviewer #2: The introduction lacks context and background information. It would be helpful to include a brief overview of the importance of perioperative pain management, the prevalence of postoperative pain, and the significance of reducing opioid usage. Additionally, mention some statistics or evidence supporting the use of multimodal analgesia.

Define all abbreviations used in the paper, especially in the abstract or introduction.

Clarify the sample size justification and whether it was calculated based on the primary outcome measures.

Expand the discussion section to interpret the results in the context of existing literature. Discuss the implications of your findings for perioperative pain management, limitations of the study, and potential future research directions.

Provide a concise and clear summary of the key findings and their implications for clinical practice or further research.

The paper contains several grammar and typographical errors. Consider proofreading the manuscript thoroughly to improve its overall quality.

6. PLOS authors have the option to publish the peer review history of their article (what does this mean?). If published, this will include your full peer review and any attached files.

Reviewer #1: No

Reviewer #2: No

---

## [Author Response · Author response to Decision Letter 0]

20 Aug 2023

We would like to thank you for your careful reading, helpful comments, and constructive suggestions, which has significantly improved the presentation of our manuscript.

 We have carefully considered all comments from the reviewers and revised our manuscript accordingly. The manuscript has also been double-checked, and the typos and grammar errors we found have been corrected. We marked out our responses to each comment from the reviewers in "Revised Manuscript with Track Changes" . We also responses to each comment in the file"Response to Reviewers". We believe that our responses have well addressed all concerns from the reviewers. We hope our revised manuscript can be accepted for publication.

---

## [Decision Letter · Decision Letter 1]

23 Oct 2023

PONE-D-23-20292R1Ultrasound-guided medial branch of the superior laryngeal nerve block to reduce peri-operative opioids dosage and accelerate patient recoveryPLOS ONE

Dear Dr. Zhang,

Thank you for submitting your manuscript to PLOS ONE. After careful consideration, we feel that it has merit but does not fully meet PLOS ONE’s publication criteria as it currently stands. Therefore, we invite you to submit a revised version of the manuscript that addresses the points raised during the review process.

=

 To make the study more clear, in the introduction, separate the primary and secondary goals. In the "materials and methods" section, move information about intervention doses and the control group to the randomization methods and rename it as "randomization." In Figure 1 (CONSORT), move it to the results section and add the numbers in each "allocated to group x." Also, clarify if blinding and allocation concealment were used. Clearly define primary and secondary outcomes, and mention any safety outcomes. Specify your statistical comparisons in advance, and address how missing data will be handled. Revise the sample size section to ensure it's clear whether Groups A and B from the pilot study are similar to this main study, rounding up the number to 18 if necessary. Finally, in Table 1, avoid performing formal tests at baseline since differences observed would be due to chance, so omit p-values

We look forward to receiving your revised manuscript.

Kind regards,

Lalit Gupta

Academic Editor

PLOS ONE

Additional Editor Comments:

This study explores how blocking the superior laryngeal nerve can reduce stress during general anesthesia. To make the study more clear, in the introduction, separate the primary and secondary goals. In the "materials and methods" section, move information about intervention doses and the control group to the randomization methods and rename it as "randomization." In Figure 1 (CONSORT), move it to the results section and add the numbers in each "allocated to group x." Also, clarify if blinding and allocation concealment were used. Clearly define primary and secondary outcomes, and mention any safety outcomes. Specify your statistical comparisons in advance, and address how missing data will be handled. Revise the sample size section to ensure it's clear whether Groups A and B from the pilot study are similar to this main study, rounding up the number to 18 if necessary. Finally, in Table 1, avoid performing formal tests at baseline since differences observed would be due to chance, so omit p-values.

Reviewers' comments:

Reviewer's Responses to Questions

**Comments to the Author**

1. If the authors have adequately addressed your comments raised in a previous round of review and you feel that this manuscript is now acceptable for publication, you may indicate that here to bypass the “Comments to the Author” section, enter your conflict of interest statement in the “Confidential to Editor” section, and submit your "Accept" recommendation.

Reviewer #2: All comments have been addressed

Reviewer #3: (No Response)

2. Is the manuscript technically sound, and do the data support the conclusions?

Reviewer #2: Yes

Reviewer #3: No

3. Has the statistical analysis been performed appropriately and rigorously? 

Reviewer #2: Yes

Reviewer #3: No

4. Have the authors made all data underlying the findings in their manuscript fully available?

Reviewer #2: Yes

Reviewer #3: Yes

5. Is the manuscript presented in an intelligible fashion and written in standard English?

Reviewer #2: No

Reviewer #3: Yes

6. Review Comments to the Author

Reviewer #2: (No Response)

Reviewer #3: This is an interesting study looking at the effect of medial branch block of the superior laryngeal acne reduce stress response in patients under going general anesthesia.

They are some fundamental points worth considering.

In the introduction section at then end clearly distinguish the primary and secondary objectives.

In the "materials and methods" section. Move the section definition of intervention dosages and control group etc… to random and control methods and actually rename that as "randomisatiom"

Figure 1 CONSORT. This needs to be in results section.

Figure 1: add the numbers in each "allocated to group x = " for all 4 groups

Mention details blinding if any and allocation concealment.

Outcomes. It's essential that outcomes are clearly defined. Both primary and secondary outcomes. Any safety outcomes?

Stats methods. Pre-specify your comparisons otherwise if multiple comparisons was multiplicity adjusted for?

Handling of missing data.

Re-write the sample size section. It's unclear whether from the pilot study Group A and B are of similar interventions as in this main study?

Round up the number to 18 as you cannot recruit 17.4 individuals.

As this is a RCT report results in line with the CONSORT guideline statement.

Table 1. As this is a randomised controlled trial, its not appropriate to perform formal tests at baseline. Any differences observed would indeed be due to chance. So omit p-values.

7. PLOS authors have the option to publish the peer review history of their article (what does this mean?). If published, this will include your full peer review and any attached files.

Reviewer #2: No

Reviewer #3: No

---

## [Author Response · Author response to Decision Letter 1]

3 Nov 2023

Dear academic editor and reviewers

We would like to thank you for your careful reading, helpful comments, and constructive suggestions, which has significantly improved the presentation of our manuscript.

We have carefully considered all comments from the reviewers and revised our manuscript accordingly. The manuscript has also been double-checked, and the typos and grammar errors we found have been corrected. All abbreviations,particularly in the abstract and introduction have been defined. In the following section,we summarize our responses to each comment from the reviewers. We believe that our responses have well addressed all concerns from the reviewers. 

In our revisions, we paid specific attention to the comments

1)“To make the study more clear, in the introduction, separate the primary and secondary goals” and“Clearly define primary and secondary outcomes, and mention any safety outcomes”

For these comment, We rewrote the last paragraph of the introduction as follows：“Our primary goal was to assess whether the medial branch block of the superior laryngeal nerve can reduce the stress response of patients undergoing general anesthesia during intubation. Our secondary goals were to assess whether the combination of other nerve blocks that can alleviate the incision pain can improve the perioperative pain of patients, thereby further reducing the dosage of opioids and speed up patient's postoperative recovery.”

2)“In the "materials and methods" section, move information about intervention doses and the control group to the randomization methods and rename it as "randomization." ” and“clarify if blinding and allocation concealment were used”

For this comment,We have made corresponding modifications according to the suggestions, and wrote a section named “Randomization and blinding”

3)“In Figure 1 (CONSORT), move it to the results section and add the numbers in each "allocated to group x." ”. 

For this comment, We added the numbers in each "allocated to group x" in Figure 1 (CONSORT). However, if we move Figure 1 (CONSORT) to the results section, it will become Figure 2. Through reading related articles published in PLOS ONE, we found that there are also articles that put Figure 1(CONSORT) in Materials and methods section. So, we have retained Figure 1 (CONSORT) in this section.

4)“Specify your statistical comparisons in advance, and address how missing data will be handled.” and “Revise the sample size section to ensure it's clear whether Groups A and B from the pilot study are similar to this main study, rounding up the number to 18 if necessary”

For this comment, We have made corresponding modifications according to the suggestions. Previously, due to a writing error, we wrote Group B instead of Group C. The largest difference was expected to be between groups A and C which are similar to this main study. The missing data were treated using the mean imputation method.

5)“avoid performing formal tests at baseline since differences observed would be due to chance, so omit p-values”

For this comment, We have made corresponding modifications according to the suggestions.

All the changes are marked out in the paper ”Revised Manuscript with Track Changes”

Thank you and best regards.

Yours sincerely,

2023.11.2

Corresponding author:

Name: Xu Zhang

E-mail: 450047830@qq.com

---

## [Decision Letter · Decision Letter 2]

16 Nov 2023

Ultrasound-guided medial branch of the superior laryngeal nerve block to reduce peri-operative opioids dosage and accelerate patient recovery

PONE-D-23-20292R2

Dear Dr. Zhang,

We’re pleased to inform you that your manuscript has been judged scientifically suitable for publication and will be formally accepted for publication once it meets all outstanding technical requirements.

Kind regards,

Lalit Gupta

Academic Editor

PLOS ONE

Additional Editor Comments (optional):

Reviewers' comments:

Reviewer's Responses to Questions

**Comments to the Author**

1. If the authors have adequately addressed your comments raised in a previous round of review and you feel that this manuscript is now acceptable for publication, you may indicate that here to bypass the “Comments to the Author” section, enter your conflict of interest statement in the “Confidential to Editor” section, and submit your "Accept" recommendation.

Reviewer #2: All comments have been addressed

Reviewer #3: All comments have been addressed

2. Is the manuscript technically sound, and do the data support the conclusions?

Reviewer #2: Yes

Reviewer #3: Yes

3. Has the statistical analysis been performed appropriately and rigorously? 

Reviewer #2: I Don't Know

Reviewer #3: Yes

4. Have the authors made all data underlying the findings in their manuscript fully available?

Reviewer #2: Yes

Reviewer #3: No

5. Is the manuscript presented in an intelligible fashion and written in standard English?

Reviewer #2: Yes

Reviewer #3: Yes

6. Review Comments to the Author

Reviewer #2: The revised manuscript greatly improves its quality, making it suitable for acceptance. The changes implemented enhance its coherence and clarity, significantly improving its overall readability and comprehension.

Reviewer #3: (No Response)

7. PLOS authors have the option to publish the peer review history of their article (what does this mean?). If published, this will include your full peer review and any attached files.

Reviewer #2: **Yes: **LALIT GUPTA

Reviewer #3: No

---

## [Editor Report · Acceptance letter]

30 Nov 2023

PONE-D-23-20292R2 

Ultrasound-guided medial branch of the superior laryngeal nerve block to reduce peri-operative opioids dosage and accelerate patient recovery 

Dear Dr. Zhang:

I'm pleased to inform you that your manuscript has been deemed suitable for publication in PLOS ONE. Congratulations! Your manuscript is now with our production department. 

Kind regards, 

on behalf of

Dr. Lalit Gupta 

Academic Editor

PLOS ONE